# Changes in the Adaptive Cellular Repertoire after Infection with Different SARS-CoV-2 VOCs in a Cohort of Vaccinated Healthcare Workers

**DOI:** 10.3390/vaccines12030230

**Published:** 2024-02-23

**Authors:** Sara Caldrer, Silvia Accordini, Cristina Mazzi, Natalia Tiberti, Michela Deiana, Andrea Matucci, Eleonora Rizzi, Stefano Tais, Fabio Filippo, Matteo Verzè, Paolo Cattaneo, Gian Paolo Chiecchi, Concetta Castilletti, Massimo Delledonne, Federico Gobbi, Chiara Piubelli

**Affiliations:** 1Department of Infectious—Tropical Diseases and Microbiology, IRCCS Sacro Cuore—Don Calabria Hospital, Negrar di Valpolicella, 37024 Verona, Italychiara.piubelli@sacrocuore.it (C.P.); 2Centre for Clinical Research, IRCCS Sacro Cuore—Don Calabria Hospital, Negrar di Valpolicella, 37024 Verona, Italy; 3Nurse Direction, IRCCS Sacro Cuore Don—Calabria Hospital, Negrar di Valpolicella, 37024 Verona, Italy; 4Medical Direction, IRCCS Sacro Cuore Don—Calabria Hospital, Negrar di Valpolicella, 37024 Verona, Italy; 5Department of Biotechnology, University of Verona, 37134 Verona, Italy

**Keywords:** SARS-CoV-2 VOCs, cell-mediated immunity, HCWs, humoral immunity, vaccine

## Abstract

Background: Currently approved vaccines are highly effective in protecting against hospitalization and severe COVID-19 infections. How pre-existing immunity responds to new variants with mutated antigens is crucial information for elucidating the functional interplay between antibodies and B and T cell responses during infection with new SARS-CoV-2 variants. Methods: In this study, we monitored the dynamics and persistence of the immune response versus different SARS-CoV-2 variants of concern that emerged during the pandemic period (2021–2022) in a cohort of vaccinated healthcare workers, who experienced breakthrough infection in the Pre-Delta, Delta, and Omicron waves. We evaluated both the humoral and cell-mediated responses after infection. We also evaluated the anti-SARS-CoV-2 antibodies levels produced by infection in comparison with those produced after vaccination. Results: Our results highlighted that the immune response against the Delta VOC mainly involved an adaptive humoral and switched memory B cells component, even 3 months after the last vaccine dose, conversely showing a high percentage of depleted adaptive T cells. Omicron infections triggered a consistent production of non-vaccine-associated anti-N antibodies, probably to balance the spike epitope immune escape mechanisms. Conclusion: Our results suggest a direct dependence between the VOC and different humoral and B and T cell balances in the post-infection period, despite the administration of a different number of vaccine doses and the elapsed time since the last vaccination.

## 1. Introduction

On 5 May 2023 the World Health Organization (WHO) declared that the spread of COVID-19 was no longer a global public health emergency; however, the WHO still recommends maintaining monitoring systems in order to avoid the spread of new variants leading to increasing cases. A deep understanding of the immune response elicited by SARS-CoV-2 breakthrough infections after previous immunization by vaccination or infections is of paramount importance to predict the future evolution of COVID-19. Despite the essential contribution of vaccines to the route out of the pandemic, it has been observed that the antibody neutralizing activity decreases over time after SARS-CoV-2 infection as well as after immunization with COVID-19 vaccines [1]. Moreover the viral capability to mutate, particularly in the spike protein sequence, continuously generates new variants able to escape the host’s antibody response. Until now, five main variants of concern (VOCs), namely Alpha, Beta, Gamma, Delta, Omicron, and descendant subvariants stand out by their ability to rapidly spread to different regions of the world [2,3], with clinical and public health significance [4]. Since the beginning of the pandemic, healthcare workers (HCWs) have played a crucial role in patients’ care, being an exposed and at-high-risk-of-infection group.

The introduction of the currently approved spike-based vaccines represented a crucial event in decreasing SARS-CoV-2 transmission through the population, and it was also a critical prophylaxis measure to increase the safety of HCWs. In particular, the SARS-CoV-2 mRNA-based vaccines were highly effective in protecting against hospitalization and severe infections; however, especially after the appearance of the Omicron variant, several breakthrough infections were observed [5,6], highlighting the need for continuous revision of the vaccines. A deeper understanding of the immune response to SARS-CoV-2 infection after vaccination and its persistence over time is a crucial point in designing the future vaccination protocols [7,8].

A recent paper demonstrated that the updated XBB.1.5-based vaccines could induce satisfactory antibody responses against the currently circulating variants, including Kraken (XBB.1.5), Pirola (BA.2.86), and Eris (EG.5.1) [9]. However, the major part of the published studies focused only on the antibody neutralizing capacity, since a major concern is represented by the reduction in vaccine efficacy against variants carrying mutations in key neutralizing spike epitopes, allowing the virus to partially evade Ab recognition. Conversely, SARS-CoV-2-specific T cell epitopes appear to be less affected by the mutations present in VOCs [10]. The T cell contribution to protection from breakthrough infections and progression to severe COVID-19 is central in the scientific debate. Moreover, it has been pointed out that the T cell response in convalescents could offer some level of protection for many months and even years [11].

Underpinning the adaptability of pre-existing (natural or vaccine-induced) immunity to new variants with mutated antigens is crucial information for elucidating the functional interplay between the B and T cell antibodies’ responses during infection with new VOCs. In this study we monitored the dynamics and persistence of the immune response versus different SARS-CoV-2 VOCs that emerged during the pandemic period, referring to the immunological state after the last vaccine dose before viral infection. We evaluated both the humoral and cell-mediated response following exposure to different SARS-CoV-2 VOCs, by assessing anti-Spike and anti-Nucleocapsid antibodies, as well as characterizing the B and T cells’ repertoire, including maturation, activation, and potential senescence or exhaustion markers, in a cohort of vaccinated HCWs who experienced breakthrough infection between March 2021 and June 2022. For each subject, we also considered the humoral response after the last vaccine dose, to evaluate the anti-SARS-CoV-2 antibodies levels produced by infection in comparison with those produced after vaccination.

## 2. Materials and Methods

### 2.1. Study Population and Sample Collection

Between January and November 2021, HCWs at the IRCCS Sacro Cuore Don Calabria Hospital (Negrar di Valpolicella, Verona, Italy) received three doses of the Pfizer-BioNTech COVID-19 monovalent spike mRNA vaccine (BNT162b2, Comirnaty^®^). HCWs volunteered to participate in a prospective longitudinal observational study to determine the dynamics and persistence of the immune response throughout the vaccination campaign. For a full description of the longitudinal study, refer to Piubelli et al. [12]. Whole blood and serum samples were collected before (pre-vaccination, PRE) and three weeks after each vaccine dose (post vaccination, PV). From March 2021 to June 2022, 35 HCWs were followed up after COVID-19 breakthrough infection with an additional blood sampling at about two weeks after SARS-CoV-2 infection (post infection, PI). To our knowledge, all the HCWs were naïve and negative for the anti-Nucleocapsid specific IgG at the moment of infection. All participants received at least two doses of the BNT162b2 vaccine before the record of breakthrough infection. The schematic timeline of the vaccination/blood sampling schedule and infection period, along with VOC dominance is reported in Appendix A. Demographic and clinical characteristics were recorded by a Clinical Research Form and are summarized in Table 1.

### 2.2. Ethics

The study protocol received ethical clearance from the Ethical Committee of Verona and Rovigo provinces (Prot n. 2916 of 19 January 2021). Participants signed an informed consent form. Samples were collected and stored in the Tropica Biobank of the IRCCS Sacro Cuore Don Calabria Hospital.

### 2.3. SARS-CoV-2 Molecular Diagnosis and VOC Assignment

SARS-CoV-2 infection was confirmed at the Department of Infectious–Tropical Diseases and Microbiology (DITM) of IRCCS Sacro Cuore Don Calabria Hospital by routine reverse-transcriptase real time PCR (RT-qPCR) on nasopharyngeal swab, according to the WHO guidelines [13]. Reverse transcription, amplification, and whole genome sequencing (WGS) of the SARS-CoV-2 genome from the swab samples were performed, as previously described [14]. Additionally, samples exhibiting ct values ≤ 30 were reverse-transcribed using the SuperScript™ VILO™ II cDNA Synthesis Kit (Invitrogen, Carlsbad, CA, USA) and amplified using the Ion AmpliSeq SARS-CoV-2 Insight Research Assay (ThermoFisher Scientific), according to the manufacturer’s instructions. After barcoding by Ion Xpress Barcode Adapters (Invitrogen) and purification with Agencourt™ Ampure™ XP beads (Beckman Coulter, Brea, CA, USA), libraries were quantified by Qubit 4 Fluorometer (Thermo Fisher Scientific, Waltham, MA, USA). The Ion Chef Instrument (Thermo Fisher Scientific) was used for clonal amplification and loading of the pooled library (30 pM) into the Ion 530 chips. WGS analysis was performed by the GeneStudio™ S5 System (Thermo Fisher Scientific). SARS-CoV-2 sequences were deposited in the GISAID open source database (https://www.gisaid.org/; accessed on 24 November 2023). Phylogenetic analysis was performed using the Nextclade sequence analysis webapp (https://clades.nextstrain.org; accessed on 24 November 2023) [15]. Clade information was described using the GISAID and Nextstrain nomenclature, and lineage information was described using the Pangolin nomenclature.

### 2.4. Specific SARS-CoV-2 Serological Tests

The serum samples collected from HCWs were tested for specific antibodies against SARS-CoV-2 antigens, in particular: IgG anti-Nucleocapsid protein (IgG-N), IgM anti-Spike protein (IgM-S) and IgG anti-receptor-binding domain (IgG-RBD-S). Antibodies were measured using automated chemiluminescent micro particle immunoassays (CMIA) with commercial kits (SARS-CoV-2 IgG, SARS-CoV-2 IgM, and SARS-CoV-2 IgG II Quant assay, Abbott, Milan, Italy) run on the ARCHITECT i2000 System (Abbott), following the manufacturer’s instructions. For IgG-N and IgM-S, the results were measured as a relative light unit (RLU) by the system optics. The RLU of the sample (S) was automatically compared with the RLU of a specific calibrator (C), resulting in an assay index (S/C). As per the manufacturer’s instructions, the interpretation of the results was as follows: IgG-N index (S/C) ≥ 1.4 = positive, IgM-S index (S/C) ≥ 1 = positive. The IgG-RBD-S results were reported as Arbitrary Unit (AU)/mL, according to the following interpretation: AU/mL ≥ 50 = positive. The AU/mL are related to the WHO binding antibody Unit/mL (BAU/mL) by the equation: BAU/mL = 0.142 × AU/mL, and the results obtained in this study have been expressed in BAU/mL.

### 2.5. Flow Cytometry Analysis of Whole Blood T and B Cell Populations

Flow cytometry analyses were performed on whole blood samples collected in EDTA and stored at −80 °C in 10% DMSO (*v*/*v*) within 6 h of the blood withdrawal. The B and T cell surface markers were tested using the Beckman Coulter DuraClone IM T cell panel (Beckman Coulter, Miami, FL, USA) and the DuraClone IM B cells panel (Beckman Coulter). Samples were prepared as recommended by the manufacturer (Beckman Coulter). To analyze the B cells subset, 300 μL of whole blood was washed and re-suspended with PBS, and 150 μL of the washed whole blood was added to reagent tube DuraClone IM B cells, vortexed, and incubated at room temperature for 20 min in the dark. Then, 2 mL of VersaLyse Solution (BD Bioscience) was added, and the samples were incubated at room temperature for 15 min, protected from direct light exposure. Samples were then washed twice with PBS, and the cell pellet was finally suspended in 200 μL of PBS before signal acquisition. Similarly, to analyze the T cells subset, 150 μL of whole blood was added to reagent tube DuraClone IM T cells, vortexed, incubated at room temperature for 20 min in the dark, and processed, as described before. Data acquisition was performed using a CytoFlex flow cytometer with the CytExpert software v2.3 (Beckman Coulter). The stopping rule was set at 10.000 events in the CD3+ gate for the T cells panel or, 1.000 events in the CD19+ gate for the B cells panel. Data concerning gated lymphocyte populations were analyzed with Kaluza software v2.1 (Beckman Coulter).

### 2.6. Statistical Analysis

Statistical analyses were performed using R software v4.2.1 (R Core Team, Vienna, Austria), and plots generated with GraphPad Prism v8.3.0 (GraphPad Software, San Diego, CA, USA). Non-parametric tests were applied according to the data distribution. Differences in cell levels were assessed using the Mann–Whitney U test or the Kruskal–Wallis test when comparing more than two groups. False discovery rate (FDR) correction was used for multiple comparisons. The Spearman coefficient was used to evaluate correlations. The significance level was set at *p* value < 0.05, and all tests were two-tailed. A multivariable linear regression model was used to investigate the effect of the VOCs on antibody concentrations, T and B cells subsets, adjusted for age and days post vaccination.

## 3. Results

### 3.1. Population Characteristics Based on the VOC

The demographic and clinical characteristics of the analyzed HCW cohort are reported in Table 1. HCWs were grouped into Pre-Delta (VOC_P_), Delta (VOC_D_), and Omicron (VOC_O_), according to the SARS-CoV-2 variant lineage causing the infection as identified by the WGS characterization of positive nasal swabs. The VOC_P_ group incorporated SARS-CoV-2 lineages circulating at the beginning of 2021, including mainly Alpha VOC and preceding VOC_D_. The identified detailed lineages or sublineages of SARS-CoV-2 and the relative incidence for each group are reported in Appendix A. As reported in Table 1, no differences in gender or age were observed among the three groups. HCWs infected with VOC_P_ or VOC_D_ had received two doses of the BNT162b2 mRNA vaccine, while those who were infected by the VOC_O_ lineages had already completed the vaccination cycle with the third dose of the vaccine (Table 1 and Appendix A). Based on the vaccination schedule and the infection period (March 2021 to June 2022), the VOC_D_ and VOC_O_ groups became infected on average more than 3 months after the last dose of the vaccine (about 6 and 4 months, respectively, Table 1); on the contrary, HCWs infected by VOC_P_ had received the last vaccine dose on average within 3 months (*p* = 0.022). Considering the clinical information, the overall symptoms were mild, with fever as the most commonly reported symptom among all HCWs, followed by nasal congestion and sore throat (Table 1 and Appendix A). Sore throat was frequently reported among VOC_O_-infected HCWs (*p* < 0.001), while anosmia and ageusia were registered mainly within the VOC_D_ group (*p* = 0.016 and *p* = 0.002, respectively).

### 3.2. The Different SARS-CoV-2 VOCs Influenced the Development of Specific Humoral Responses

Serological features, consisting of IgM-S, IgG-N, and IgG-RBD-S antibody levels, were evaluated before vaccination (PRE_1_ and PRE_3_), three weeks after the last dose of vaccine (PV), and after the SARS-CoV-2 infection (PI), as shown in Appendix A. Analysis of the IgM-S revealed low levels of this type of antibody without any statistically relevant difference among groups, as reported in Figure 1a and Table 2. The observed low production of IgM-S is in line with other literature data reporting that a proportion of patients never develop IgM [12,16,17,18]. Comparing the two time points (PI versus PV), as expected, after infection, all groups seroconverted for the specific anti-Nucleocapsid protein; despite this, a higher increase in the IgG-N level was observed after VOC_O_ breakthrough infection (*p* = 0.008, Figure 1b). Comparing the different infection groups, we observed that the VOC_D_ and VOC_O_ groups produced higher levels of post-infection IgG-RBD-S compared to the VOC_P_ group (Table 2 and Figure 1c; PI values: VOC_P_ median 1108 BAU/mL; VOC_D_ median 6809 BAU/mL; *p* = 0.006; VOC_O_ median 3832 BAU/mL; *p* = 0.016). Interestingly, we observed that only the VOC_D_ group had a significant rise in PI IgG-RBD-S antibody levels, as compared to PV (*p* = 0.033, Figure 1c). Moreover, as previously noted [19], 90 days after the second dose, the titer of IgG-RBD-S antibodies produced by vaccination decreased. Due to this evidence, we stratified the PI results according to the time from the last vaccine dose (before and after 90 days), to evaluate a possible influence of the time lapse between vaccination and infection on the antibody levels. Considering the distinct VOCs, we detected a lower production of circulating IgG-RBD-S antibody levels only in HCWs infected by VOC_O_ and vaccinated more than 90 days before (*p* = 0.025, Figure 1d). This lower level was also observed in the other two groups but without statistical significance, probably due to the low number of subjects stratified in each group.

### 3.3. The Frequency of B Cells Was in Accordance with the Anti-RBD-S IgG Response and VOC

In addition to SARS-CoV-2-specific antibodies, the maturation stages of circulating B cells were also analyzed. In particular, B memory cells were evaluated to clarify the role of the adaptive humoral response in long-lasting immunity. The frequency of all B cell subsets is described in Table 3. As described in Figure 2a, the measured frequency of total B cells (CD19+) was significantly higher in the HCWs infected with VOC_D_ (median = 10.4%, *p* = 0.023) and VOC_O_ (median = 8.2%, *p* = 0.023), compared to those with VOC_P_ (median = 4.8%). Correlating the percentage of B cells with the IgG-RBD-S levels (Figure 2b) among the three groups, despite the lower IgG-RBD-S production observed after VOC_O_ infection, we observed a similar frequency of B cells between the VOC_D_ and VOC_O_ groups, while lower IgG-RBD-S levels and B cell frequencies were observed in the VOC_P_ group. Moreover, we stratified the PI results according to the time from the last vaccine dose (before and after 90 days), to evaluate a possible influence of the time lapse between vaccination and infection on the B cell levels. Considering the distinct VOCs, we detected higher B cell levels in HCWs infected with VOC_D_ both before and after 90 days post vaccination, as represented in Figure 2c and Table 3).

Switched B cells levels remained balanced among groups within 90 days post vaccination; however, after 90 days, HCWs infected by VOC_D_ (median = 59.1%) showed a higher level than the other groups (median = 43%; *p* = 0.02) (Figure 2c, Table 3). Finally, we evaluated the difference in B cell subtype frequencies between the three groups of HCWs with reference to the VOC_D_ patients, also normalizing by age and days elapsed since vaccination (Figure 2d, and Appendix A). This analysis confirmed that total B and switched B cell levels were on average higher in the VOC_D_ group, strengthening our above observation. Interestingly, we observed a median increase of about 8 times of the transitional B cell (TrB, CD27- CD38^high^ CD24^high^) levels only in the HCWs affected by VOC_O_, compared to those with VOC_D_ (Figure 2d).

### 3.4. The Immunophenotype of the T Lymphocytes Compartment Correlated with Different VOCs

The differences in the T cells’ maturation subset frequencies among the three groups of HCWs based on VOC breakthrough infection are described in Table 4. At a glance, the VOC_P_ HCWs showed a lower relative frequency of CD3+ T lymphocytes (8%), compared to those with VOC_D_ (median 25%, *p* = 0.043) or VOC_O_ (median 21%) infection. Moreover, the frequency of both T helper CD4+ (TH) and cytotoxic T CD8+ (CTL) lymphocytes was different in VOC_P_ breakthrough infections, with TH less represented in VOC_P_ infections (36% vs. 52%, *p* = 0.055 and 54%, *p* = 0.045) and CTL more represented (VOC_P_ median 50%) than in the other two groups (both VOC_D_ and VOC_O_ median = 35%, *p* = 0.038).

Using surface marker staining, we assessed the frequencies of the T cell maturation stage distribution of naïve T cells (TN, CD45RA+ CCR7+), central memory T cells (TCM CD45RA- CCR7+), effector memory T cells (TEM CD45RA- CCR7-), and effector memory-expressing CD45RA T cells (TEMRA CD45RA+ CCR7-) in both the CTL and TH populations. The CTL descriptive analysis indicated a lower TEMRA-CD8+ relative frequency in the HCWs infected by VOC_O_ (median 14%), with a statistically significant difference compared to the VOC_P_ group (median 25%, *p* = 0.054), as presented in Figure 3a and Table 4. Therefore, we examined the exhausted or senescent phenotype of T cells, by measuring, respectively, the Programmed cell death protein 1 (PD-1) and CD57 expression on the cells’ surface (Figure 3b). Regarding the total CTL, we observed a higher level of both exhausted and senescent phenotype (CD57+/PD1+) in HCWs infected by the VOC_P_ (median 14%), with statistical significance, compared to the VOC_O_ group (median 7%, *p* = 0.039). In particular, the VOC_O_ group presented the lowest percentage of TEM-CD8+ (CD57+/PD1+) cells (median 4%), as compared to both the VOC_P_ (median 21%, *p* = 0.001) and VOC_D_ (median 12%, *p* = 0.006) groups, as reported in Figure 3c and Table 4.

In agreement, a more efficient and proliferative population of TEM-CD8+ (CD57-/PD1-) cells was prominent in HCWs with the VOC_O_ breakthrough infection (median 53%), compared to the other two groups (VOC_P_ median 31%, VOC_D_ median 35%). As for TEM-CD8+, a similar trend could be observed by considering the relative frequencies of the TEMRA-CD8+ cells among the three groups of HCWs (Figure 3d). A significant increase in exhausted and senescent TEMRA-CD8+ (CD57+/PD1+) levels was shown in HCWs after VOC_D_ challenge (median 14%) compared with VOC_O_ (median 3%; *p* = 0.024); on the other hand, the VOC_O_ group revealed higher levels of TEMRA-CD8+ CD57-/PD1- (median 35%) compared to the VOC_P_ group (median 13%, *p* = 0.05, Figure 3d and Table 3).

### 3.5. The T Helper Lymphocytes Compartment Was Correlated with the VOC and Also with the Time Post Vaccination

Considering the maturation state of the TH cells (Figure 4a), we observed a significantly lower frequency of TN-CD4+ in the VOC_P_ group (median 10%), compared to both the VOC_D_ and VOC_O_ groups (*p* = 0.031 and *p* = 0.001, respectively, Table 4 and Figure 4a). Higher levels of TCM-CD4+ were observed in the VOC_D_ group, compared to the VOC_O_ group (*p* = 0.033). Furthermore, differences were observed by analyzing the subgroups of TH lymphocytes, as shown in Figure 4b. Evaluating the senescence and exhaustion state, we observed a significant amount of functional Th cells (CD4+ CD57-/PD1-) in HCWs affected by the VOC_O_ (median 88%), as compared to the VOC_D_ (59%; *p* = 0.003) or VOC_P_ (67%; *p* = 0.045, Table 4) groups. A high percentage of exhausted cells (CD57-/PD1+) were identified in HCWs infected by the VOC_D_, compared to VOC_P_ and VOC_O_ (*p* = 0.014 and *p* < 0.001, respectively), with the VOC_O_ group presenting a lower level (Figure 4b). In particular, regarding the TEM-CD4+ we observed a lower percentage of active (CD57-/PD1-) cells in the VOC_D_ group (*p* = 0.021 and *p* = 0.002) versus VOC_P_ and VOC_O_, respectively, while a higher component of exhausted TEM-CD4+ (CD57-/PD1+) was observed in the VOC_D_ group (median 52%) than in the VOC_O_ group (median 8%, *p* < 0.001; Table 4 and Figure 4c). Considering the relative frequency of TCM-CD4+, we observed a relevant presence of functional cells (CD57-/PD1-) in all three groups, despite a significant amount of senescent CD57+ cells in HCWs with VOC_D_ compared to the VOC_P_ group (*p* = 0.001, Table 4 and Figure 4d).

In addition, we performed a multivariable linear regression analysis in order to evaluate the difference in T cell subtype frequencies between the three groups, taking the VOC_D_ -infected subjects as a reference and normalizing by age and days elapsed since the last vaccination dose (Figure 5a and Appendix A). This analysis confirmed that, on average, HCWs with the VOC_O_ infection had about 30% more functional non-senescent and non-exhausted TEM-CD4+. Moreover, both the VOC_O_ and VOC_P_ groups presented lower levels of exhausted TEM-CD4+ (CD57-/PD1+) than the VOC_D_ group, as presented in Figure 5a. Also in the CTL compartment, we observed a prevalence of non-senescent and non-exhausted TEM-CD8+ and TEMRA-CD8+ (CD57-/PD1-) in HCWs infected by VOC_O_, compared to those with VOC_D_ (Figure 5a). Finally, we also evaluated whether this difference in TEM-CD4+ was due to the time lapse between the last vaccination dose and the SARS-CoV-2 infection (Appendix A and Figure 5b). Through this evaluation, we confirmed that TEM-CD4+ (CD57-/PD1-) cells were predominant in HCWs affected by the VOC_O_ variant, both before (74%) and, significantly, after 90 days from vaccination (62%, *p* = 0.04), compared to those with VOC_D_. In contrast, a high number of exhausted TEM-CD4+ (CD57-/PD1+) cells were retrieved in the VOC_D_ group (median 52%), compared to the VOC_O_ group (median 8%), particularly when infection occurred after 90 days from vaccination (*p* = 0.023, Figure 5b).

## 4. Discussion

The rapid development of different COVID-19 vaccines led to the formal ending of the pandemic in 2023. Vaccination was the key solution to achieving a substantial reduction in the number of new daily infections, the frequency of hospitalizations, and the fatal events, but it did not guarantee long-term protection from new symptomatic infections [20,21], especially considering the advent of new variants [22]. Protection against SARS-CoV-2 infection and symptomatic and severe COVID-19 disease is driven by different humoral and cellular components of the immune system over time [23], whose modulation and activation are strongly influenced by vaccination.

In our previous study, we deeply analyzed the humoral response to the Comirnaty vaccine in a large population of vaccinated HCWs at IRCCS Sacro Cuore Don Calabria Hospital, longitudinally followed up until 6 months after the third dose [12]. In the present study, we focused our analysis on a subgroup of this longitudinal cohort, i.e., 35 vaccinated subjects who experienced SARS-CoV-2 breakthrough infection between March 2021 and June 2022, thus after two or three vaccine doses and during different VOC waves (VOC_P_, VOC_D_, and VOC_O_). In these subjects, we evaluated both the humoral and adaptive cell-mediated immune response after infection, at the moment of their return to work, thus about 10.5 days on average (Table 1) after the diagnosis. The aim of the study was to investigate how the immune response evolved, considering the number of vaccine doses, the time elapsed from the last dose, and the VOC responsible for the breakthrough infection. The reported symptoms were generally mild, with anosmia, ageusia, and fever frequently reported among HCWs with VOC_D_ infection, while sore throat, cough, fever, and nasal congestion were registered mainly within the VOC_O_ group. On average, infection occurred 4 months after the last vaccine dose (Table 1).

For all subjects, the IgG-RBD-S antibody response was activated by the infection, reaching at least the PV level (Figure 1c). Among the three different infection groups, the VOC_P_ produced the lowest levels of post-infection IgG-RBD-S, probably due to the similarity between the S protein of the pre-Delta variants and the Wuhan-based vaccine. This could result in the production of higher affinity antibodies for the pre-Delta variant, which could achieve a more efficient response, even with lower levels. Interestingly, the highest IgG-RBD-S antibody response was recorded among the VOC_D_ group, with a statistically significant increase compared to the PV level (*p* = 0.033).

As suggested by the analysis considering the time from the last vaccination dose (Figure 1d), HCWs infected with VOC_D_ generated a higher IgG-RBD-S antibody response despite having received the last vaccine dose more than 90 days before (median 6 months). These observations reasonably suggest that the immune response triggered by the VOC_D_ virus could mainly involve the humoral component of the adaptive system in a very sustained and continuous way over time.

Anti-N IgG antibodies were measured in all infected individuals, but the immune response against the VOC_O_ virus, in addition to an effective IgG-RBD-S response, also triggered a significant increase (*p* = 0.008) in this non-vaccine-associated antibody response. This observation suggests that high levels of IgG-N could be elicited by the host, in order to compensate for the spike epitope immune escape mechanisms adopted by VOC_O_, which may reduce the neutralizing effect of the memory humoral compartment elicited by a monovalent vaccine [24,25,26].

This evidence could suggest a direct dependence between the VOC and the different antibody production in the post-infection period, despite the administration of a different number of vaccine doses and the elapsed time since the last vaccination.

Accordingly, in VOC_D_ -infected HCWs, we observed a correlation between the IgG-RBD-S antibody levels and the total and switched B cells frequency, despite having received the last vaccine dose more than 90 days before. These data suggest the synergy between the B cell frequencies and the IgG-RBD-S antibody production was preserved, especially after VOC_D_ breakthrough, which may provide a good indication of vaccine efficacy over time [27]. In contrast, high levels of peripheral transitional B cells were observed only in HCWs infected by VOC_O_. Their role in the SARS-CoV-2 response was already highlighted in the literature: they were found at higher levels in COVID-19 patients presenting mild symptoms, while they have been shown to be absent in patients with severe symptoms [28,29]. These cells, typically defined as “immature”, represent a crucial link between the B and T compartments, by directing bone marrow immature B cells to mature peripheral B cells, by regulating CD4+ T cells proliferation and differentiation toward TH effector cells [30,31], and overall, by activating TH cells.

As previously observed, after primary infection, both symptomatic and asymptomatic COVID-19 patients generate a robust CD4+ and CD8+ memory T cell response, in parallel with the humoral response [32]. In adaptive immunity, CTL plays an essential role in controlling viral infection by killing virus-infected cells and producing effector cytokines [33]. In COVID-19 patients, the CD8+ T cell population undergoes important quantitative and qualitative changes. As already described in virus-infected immunocompetent patients, as well as in our cohort of non-hospitalized subjects, the phenotypic pattern of CTL showed an increase in TEM-CD8 cells, reflecting the ongoing immune activation (Figure 3a). The co-inhibitory receptor PD-1 plays an important immune regulatory role by reducing initial T cell activation, fine-tuning T cell differentiation and effector functions, and contributing to the development of immunological memory [34]. In our cohort, we found a significant increase in exhausted (PD1+) and senescent (CD57+) TEMRA-CD8+ levels in HCWs after VOC_D_ infection compared to those with VOC_O_ (Figure 3d). This latter group instead revealed an overall efficient and vital CTL phenotype for both TEMCD8+ and TEMRACD8+ (CD57-/PD1-, Figure 3c,d). Moreover, by normalizing the data by age and days elapsed since vaccination, we confirmed that HCWs with a VOC_O_ infection had about 20% more functional non-senescent and non-exhausted TEMCD8+ (Figure 5a) and about 20% less senescent TEMCD8+ (Figure 5b) cells, compared to HCWs with VOC_D_. These data indicate the presence of a more active CTL compartment in VOC_O_ -infected patients.

As widely agreed, the TH cells play essential roles in coordinating immune responses through the induction of B cells for neutralizing antibodies’ production, as well as promoting both the effector activity of CTL cells and the establishment of memory B and T cell compartments [35,36]. The TCM-CD4+ cells are directly involved in supporting antibody production: once the antigen has been presented by the B lymphocyte, only in presence of an adequate stimulus of T CD4+ lymphocytes can the development of immunoglobulins begin [37]. Considering the maturation stages and the senescence and exhaustion markers of TH cells, we observed differences in the balances among vaccinated HCWs infected by different VOCs. In the VOC_D_ group, we observed higher levels of TCM-CD4+, suggesting the presence of a tangible reservoir of memory cells after the SARS-CoV-2 infection, and a high amount of exhausted TEM-CD4+ cells, particularly when infection occurred after 90 days from vaccination (*p* = 0.023, Figure 5b). Conversely, we detected a significant and functional component of TEM-CD4+ in subjects with VOC_O_ breakthrough infection, indicating that the T cell-mediated response represents the relevant fightback against the virus. These differences between the two groups were also evaluated in association with the elapsed time between the last vaccination dose and the SARS-CoV-2 infection, with results confirming the discrepancy observed in the total number of subjects in both groups.

The main limitation of the study is the lack of antigen-specific T cell activation experiments to confirm our results. This was due to the storage of blood samples in DMSO, which limited our testing capabilities. Further studies should be conducted to evaluate the T and B cells’ in vitro response after a specific stimulus.

## 5. Conclusions

In conclusion, our data evidenced that, in subjects immunized with monovalent vaccine, thus designed on the Wuhan original sequence, the immune response is strictly dependent on the VOC. The Delta lineage infection triggered an efficient adaptive humoral response characterized by higher vaccine-associated antibody production, a functional B cell maturation over time, and a less viable T cell compartment. In contrast, the Omicron lineage infection, while triggering an efficient T cell-mediated response, underscoring a lower sensitivity of the T cell response to mutations present in VOC_O_ [19,38,39], even long after the last vaccination dose, resulted in a greater non-vaccine-associated antibody response strongly encouraging the continuous updating of the vaccine formulation.

The balancing of the host immune response to achieve viral clearance is influenced by both the SARS-CoV-2 escape mechanisms and the immune response also conditioned by the original “antigenic sin”. These mechanisms become a crucial aspect in defining the magnitude of the immune response to new VOCs, the risk of reinfection, and most importantly, in the development of new vaccines aimed at effectively managing viral infections, especially in individuals with limited immunological capability.

## Figures and Tables

**Figure 1 vaccines-12-00230-f001:**
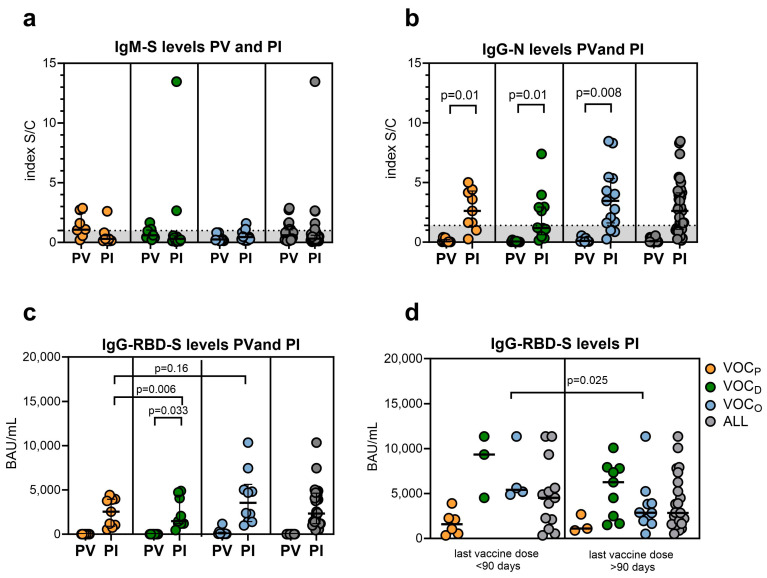
SARS-CoV-2 antibody levels. Scattered plot representing: (**a**) the different levels of IgM-S post vaccination (PV) and after SARS-CoV-2 infection (PI) separated by VOC; (**b**) the different levels of IgG-N PV and PI separated by VOC; (**c**) the different levels of SARS-CoV-2 IgG-RBD-S PV and PI, after grouping for different SARS-CoV-2 VOCs. (**d**) Scattered plot representing the IgG-RBD-S measured PI considering the timing after the last vaccination dose (within or after 90 days). Statistical significance set at *p* value < 0.05 was assessed using the Mann–Whitney U test. FDR correction was used for multiple comparisons. Dots represent individual observations; the bold line on each box indicates the median and IQR.

**Figure 2 vaccines-12-00230-f002:**
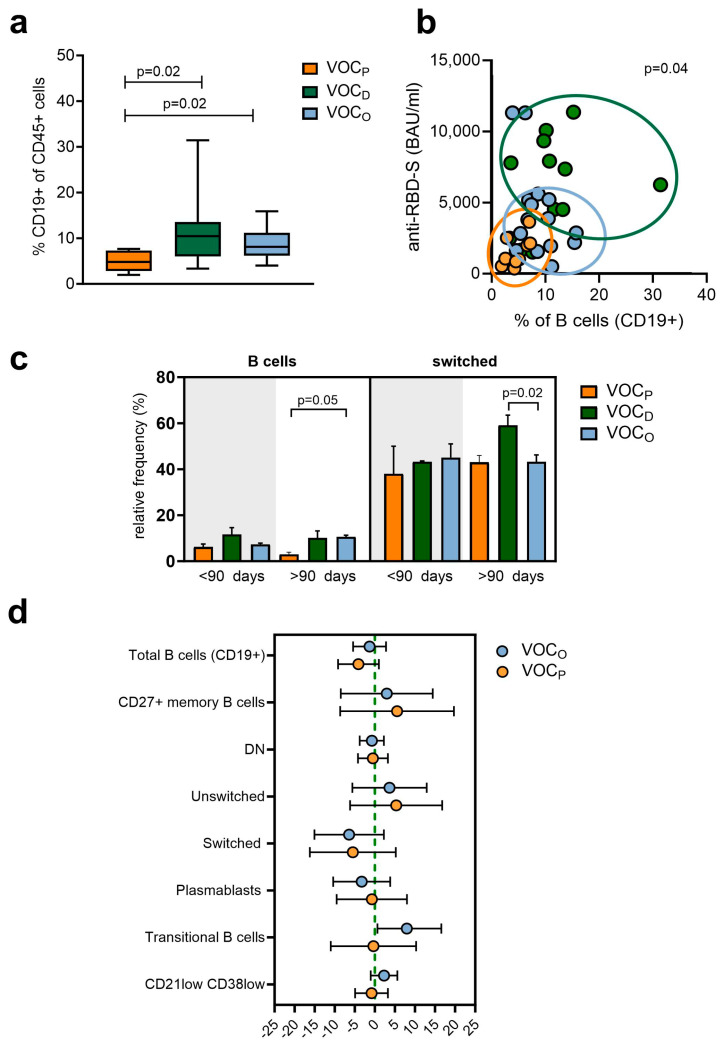
Subpopulation B cell frequencies according to VOCs. (**a**) Scatter plot representing total B cell (CD45+/CD19+) frequencies depending on the VOC. Box and whiskers bar represent median (central bar) ± 95% confidence intervals (upper and lower bars). Statistical analysis by two-sided Mann–Whitney nonparametric test; if not indicated, *p* value is not significant. FDR correction was used for multiple comparisons. (**b**) Correlation between anti-RBD antibody levels and the total B levels according to the VOC. Dots represent individual observations: orange dots for VOC_P_, green dots for VOC_D_ and light blue for VOC_O_. The distribution was evaluated by the Spearman’s rank correlation Rho. Rho value = 0.344 (low correlation); *p* = 0.04 (**c**) Representative bar plot of total B (left graph) and switched B cells levels (right graph) depending on the VOC and elapsed time from the last vaccine dose. Statistical significance was assessed using the Mann–Whitney U test. FDR correction was used for multiple comparisons. Data represent the median with error bars. (**d**) Forest plot representing the impact of different VOCs on the B cell subpopulations of HCWs adjusted by age and days post vaccination. VOC_D_ was used as the reference (green line). Dots represent the models’ coefficients ± 95% confidence intervals (95% CI).

**Figure 3 vaccines-12-00230-f003:**
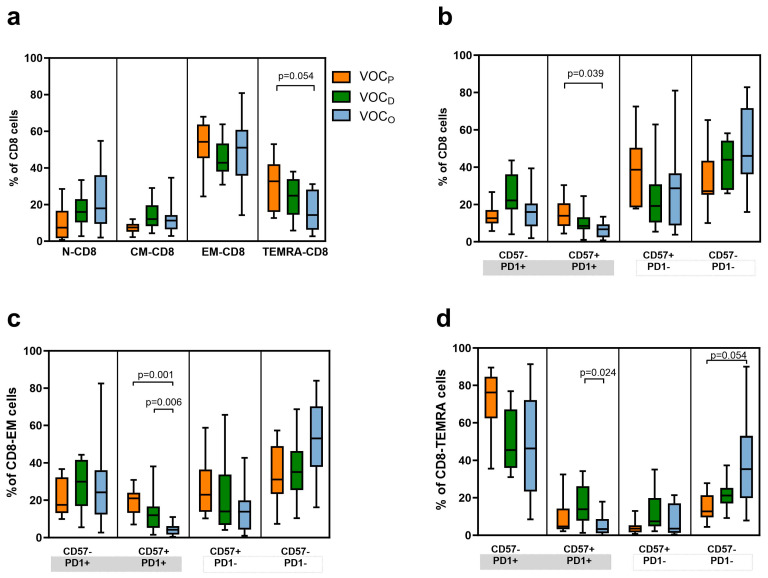
Maturation and exhaustion of CTL cell frequencies according to the VOC. Representative box and whiskers chart of all subtypes of CTL cells. T cells are identified as: naïve (TN) expressing CCR7 + CD45RA + CD28 + CD27+ cells; central memory (TCM) CCR7 -CD45RA + CD28 + CD27±; effector memory (TEM) CCR7 -CD45RA -CD28 ± CD27±; effector memory-expressing CD45RA (TEMRA) CCR7 - CD45RA + CD28 - CD27-. Moreover markers for senescence (CD57) and exhaustion were evaluated (PD-1). (**a**) Box and whiskers chart of CTL (CD8+) subtypes based on their maturation status. Data represent the median (central bar) ± 95% confidence intervals. (**b**) Representative box and whiskers chart of all CTLs based on markers for senescence (CD57) and exhaustion (PD-1) depending on VOC subtype. Data represent the median (central bar) ± 95% confidence intervals. (**c**) Representative box and whiskers chart for PD-1 and CD57 expression for T_EM_-CD8 cells or TEMRA-CD8+ (**d**) depending on VOC. Data represent the median (central bar) ± 95% confidence intervals. Multiple comparison analysis was performed to compare values between the three groups based on VOCs, and the *p* values shown were obtained by pairwise comparisons with FDR correction.

**Figure 4 vaccines-12-00230-f004:**
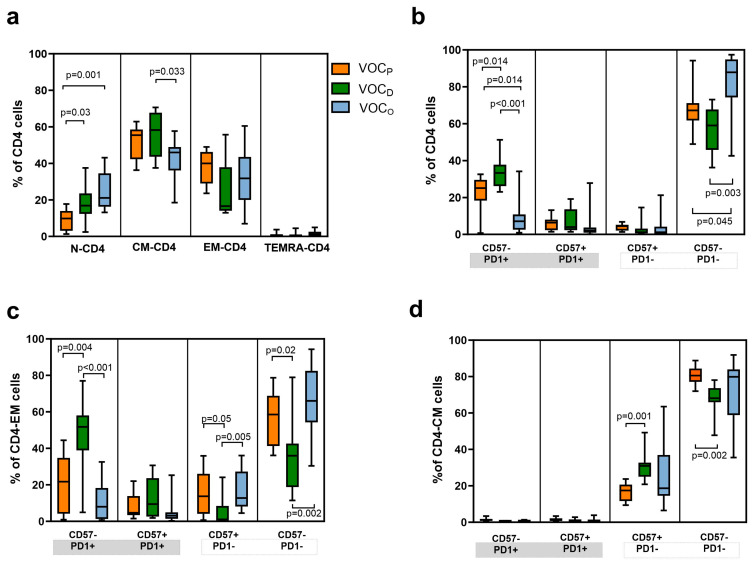
Maturation and exhaustion of T helper cells frequencies according to the VOC. Representative box and whiskers chart of all subtypes of T helper (CD4+) cells. T cells are identified as naïve (TN), central memory (TCM), effector memory (TEM), and effector memory-expressing CD45RA (TEMRA). Moreover, markers for senescence (CD57) and exhaustion were evaluated (PD-1). (**a**) Box and whiskers chart of T helper subtypes based on their maturation status. Data represent the median (central bar) ± 95% confidence intervals. (**b**) Representative box and whiskers chart of all T helper cells based on markers for senescence (CD57) and exhaustion (PD-1) depending on the VOC subtype. Data represent the median (central bar) ± 95% confidence intervals. (**c**) Representative box and whiskers chart for PD-1 and CD57 expression for TEM-CD4 and TCM-CD4 (**d**), depending on the VOC. Data represent the median (central bar) ± 95% confidence intervals. Multiple comparison analysis was performed to compare values between the three groups based on the VOC, and the *p* values shown were obtained by pairwise comparisons with FDR correction.

**Figure 5 vaccines-12-00230-f005:**
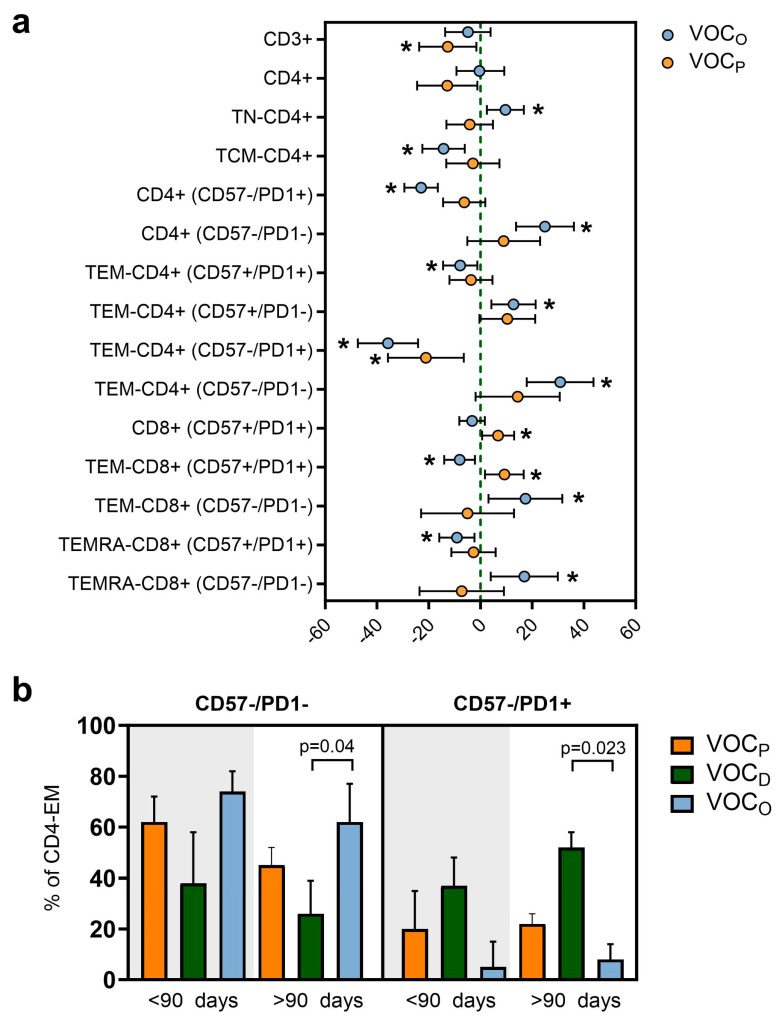
T cell floating levels regarding the vaccination time elapsed and the VOC. (**a**) Forest plot representing the impact of the type of VOC on different T cell subpopulations of HCWs adjusted by age and days post vaccination. The VOC_D_ variant was used as reference. Dots represent the models’ coefficients ± 95% confidence intervals (95% CI). Only significant results are expressed in the graph and indicated with *. (**b**) Representative bar plot of non-exhausted and non-senescent T_EM_-CD4 cell levels (left graph) or exhausted ones (right graph) depending on the VOC and the elapsed time from the last vaccine dose. Statistical significance was assessed using the Mann–Whitney U test. FDR correction was used for multiple comparisons. Data represent the median with error bars.

**Table 1 vaccines-12-00230-t001:** Baseline demographic and clinical characteristics of HCWs included in the study with regard to the type of VOC at infection.

	Variants of Concern				
	VOC_P_ (n = 9)	(n = 12)	VOC_O_ (n = 14)	All (n = 35)	*p* Value ^2^
Gender (F), n (%)	3 (33.3%)	7 (58.3%)	8 (53%)	18 (50%)	0.497 ^§^
Age (years), median [IQR]	46 [32–58]	45 [36–55]	48 [39–51]	47 [23–73]	0.912 ^‡^
No. doses of vaccine, (n)	2 (9)	2 (11), 3 (1)	3 (14)	2 (20), 3 (15)	***p* < 0.001 ^‡^**
Subjects infected after 90 days from last vaccine dose; n (%)	3 (33.3%)	9 (75%)	10 (71.4%)	22 (62.8%)	0.110 ^§^
Delay between positive molecular swab and last vaccine dose (months), median [IQR]	2.5 [2.5–3.3]	6.3 [3.6–7.5]	3.8 [2.9–6.1]	3.9 [1.3–9.5]	**0.022 ^‡^**
Delay between positive and negative molecular swab (days), median [IQR]	12 [5–20]	13.5 [3–21]	10 [6–16]	10.5 [3–21]	0.095 ^‡^
Type of symptoms, n (%) No symptoms	3 (33.33%)	-	1 (7.10%)	4 (11.42%)	0.070 ^§^
Fever	3 (33.33%)	6 (50.00%)	6 (42.90%)	15 (42.85%)	0.824 ^§^
Anosmia	2 (22.22%)	7 (58.33%)	1 (7.10%)	10 (28.57%)	**0.016 ^§^**
Sore throat	-	2 (16.66%)	11 (78.60%)	13 (37.14%)	**<0.001 ^§^**
Ageusia	2 (22.22%)	7 (58.30%)	-	9 (25.71%)	**0.002 ^§^**
Cough	-	5 (41.66%)	6 (42.90%)	11 (31.42%)	0.065 ^§^
Myalgia	3 (33.33%)	4 (33.33%)	4 (28.60%)	11 (31.42%)	1.000 ^§^
Intestinal disorder	1 (11.11%)	2 (16.66%)	-	3 (8.75%)	0.330 ^§^
Headache	2 (22.22%)	1 (8.3%)	4 (28.60%)	7 (20.00%)	0.499 ^§^
Nasal congestion	3 (33.33%)	6 (50.00%)	5 (35.70%)	14 (40.00%)	0.737 ^§^
Chest pain	1 (11.11%)	-	-	1 (2.85%)	0.251 ^§^

Categorical variables are expressed as n/N (%). Continuous variables are expressed as the median (IQR). ^§^ chi-square test. ^‡^ Wilcoxon–Mann–Whitney test. ^2^ Fisher’s exact test; Kruskal–Wallis rank sum test. VOC_P_, VOC_D_, and VOC_O_ indicate the groups of pre-Delta, Delta, and Omicron lineages, respectively.

**Table 2 vaccines-12-00230-t002:** Antibody levels stratified for SARS-CoV-2 VOCs and time of infection.

	SARS-CoV-2 VOCs					Multiple Comparisons’ *p* Value ^§^		
	VOC_P_	VOC_D_	VOC_O_	All	*p* Value ^2^	VOC_D_ vs. VOC_O_	VOC_D_ vs. VOC_P_	VOC_O_ vs. VOC_P_
**Antibody levels**								
Post-Vaccination (PV) subjects	n = 9	n = 9	n = 10	n = 28				
IgM-S (index S/C) ^1^	1.09 [0.99–2.73]	0.60 [0.43–0.96]	0.26 [0.19–0.70]	0.71 [0.26–1.10]	0.021	0.099	0.112	0.06
IgG-N (index S/C) ^1^	0.04 [0.02–0.07]	0.03 [0.02–0.08]	**0.12 [0.06–0.36] ^✝^**	0.06 [0.03–0.15]	0.038	0.059	0.722	0.09
IgG-RBD-S (BAU/mL) ^1^	2547 [1045–3911]	**1477 [1199–4068] ^✝^**	3540 [1646–4970]	2336 [1198–4495]	0.215	0.23	0.659	0.23
Post-Infection (PI) subjects	n = 9	n = 12	n = 14	n = 35				
IgM-S (index S/C) ^1^	0.30 [0.14–0.31]	0.31 [0.12–0.52]	0.45 [0.28–0.68]	0.32 [0.06–26.22]	0.315	0.455	0.972	0.416
IgG-N (index S/C) ^1^	2.62 [1.63–4.16]	1.19 [0.92–2.93]	**3.11 [1.63–4.23] ^✝^**	2.6 [0.15–40.15]	0.321	0.472	0.479	0.777
IgG-RBD-S (BAU/mL) ^1^	1108 [773–2401]	**6809 [3334–8749] ^✝^**	3869 [2372–5233]	3832 [1806–5945]	**0.003**	0.226	**0.006**	**0.016**
Delay between infection and last vaccine dose								
PV < 90 days	n = 6	n = 3	n = 3	n = 12				
IgM-S (index S/C) ^1^	1.06 [0.68–2.32]	0.60 [0.28–0.90]	0.32 [0.23–0.57]	0.71 [0.30–1.05]	0.236	0.663	0.549	0.467
IgG-N (index S/C) ^1^	**0.06 [0.04–0.29] ^✝^**	0.08 [0.02–0.09]	0.37 [0.21–0.46]	0.08 [0.04–0.36]	0.374	0.574	0.897	0.574
IgG-RBD-S (BAU/mL) ^1^	2552 [733–4012]	2060 [1627–3064]	5022 [3654–6241]	3099 [1194–4158]	0.203	0.286	0.699	0.286
PV > 90 days	n = 3	n = 9	n = 10	n = 22				
IgM-S (index S/C) ^1^	1.61 [1.3–2.2]	0.76 [0.42–1.15]	0.3 [0.2–0.5]	0.7 [0.3–1.2]	0.069	0.156	0.156	0.156
IgG-N (index S/C) ^1^	**0.02 [0.02–0.03] ^✝^**	0.03 [0.02–0.04]	0.12 [0.06–1.25]	0.04 [0.03–0.13]	**0.025**	0.078	0.354	0.078
IgG-RBD-S (BAU/mL) ^1^	2547 [1295–3561]	1349 [1138–4742]	3540 [1410–4994]	2386 [1221–4728]	0.785	1.00	1.00	1.00
PI < 90 days	n = 6	n = 3	n = 3	n = 12				
IgM-S (index S/C) ^1^	0.31 [0.18–0.69]	0.23 [0.14–1.45]	0.52 [0.30–0.92]	0.31 [0.2–0.5]	0.826	1.00	1.00	1.00
IgG-N (index S/C) ^1^	2.62 [0.94–4.18]	1.09 [0.57–2.62]	3.91 [2.08–4.79]	2.92 [1.00–4.03]	0.390	0.594	0.594	0.594
IgG-RBD-S (BAU/mL) ^1^	1581 [666–2226]	9341 [6936–10,351]	**5416 [5124–7062] ^✝^**	4531 [2117–5629]	**0.011**	**0.042**	1.00	**0.042**
PI > 90 days	n = 3	n = 9	n = 10	n = 22				
IgM-S (index S/C) ^1^	0.2 [0.13–0.25]	0.4 [0.12–0.52]	0.5 [0.3–0.6]	0.31 [0.17–0.81]	0.236	0.579	0.579	0.227
IgG-N (index S/C) ^1^	2.62 [2.12–3.82]	1.21 [1.07–2.95]	2.42 [1.63–4.09]	2.35 [1.18–3.84]	0.55	0.607	0.607	0.933
IgG-RBD-S (BAU/mL) ^1^	1108 [999–1894]	6261 [2494–7797]	**2866 [2022–3887] ^✝^**	2866 [1730–6006]	0.091	0.236	0.193	0.226

The antibody value is expressed depending on the index (index); ^1^ continuous variables are expressed as the median [interquartile range, IQR]. ^2^ Kruskal–Wallis rank sum test. ^§^ Wilcoxon rank sum test with FDR correction. ^✝^ Wilcoxon–Mann–Whitney test for paired data.

**Table 3 vaccines-12-00230-t003:** B cell subtype frequencies according to VOCs.

	VOCs			Multiple Comparisons’ *p* Value ^3^			
Characteristic	VOC_P_	VOC_D_	VOC_O_	VOC_D_ vs. VOC_O_	VOC_D_ vs. VOC_P_	VOC_O_ vs. VOC_P_	*p* Value ^2^
	n = 9 ^1^	n = 12 ^1^	n = 14 ^1^				
Leukocytes (CD45+)	17 [14–22]	23 [3–34]	14 [9–26]	0.86	0.86	0.86	0.81
Total B cells (CD19+)	4.8 [3.0–7.1]	10.4 [7.1–13.3]	8.2 [6.5–11.1]	0.55	**0.02**	**0.02**	**0.01**
CD27-- (CD19+)	63 [47–70]	67 [59–76]	60 [55–71]	0.44	0.44	0.72	0.42
CD27+ memory B cells	37 [30–52]	33 [24–40]	39 [25–44]	0.51	0.51	0.48	0.48
Switched (CD27+ IgD-IgM-)	39 [36–49]	53 [44–62]	43 [37–49]	0.08	0.08	0.73	0.06
Unswitched (CD27+ IgD+ IgM+)	59 [48–63]	46 [38–57]	55 [46–58]	0.39	0.29	0.39	0.20
DN (CD27- IgD- IgM-)	3.55 [1.81–6.84]	3.76 [2.40–7.84]	3.65 [2.51–5.28]	0.87	0.86	0.86	0.87
Naïve B cells (CD27- IgD+)	62 [40–70]	61 [49–74]	59 [50–70]	0.94	0.94	0.94	0.78
Plasmablasts (Switched B cells CD27^high^ CD38^high^)	8 [0–11]	12 [7–18]	6 [4–11]	0.21	0.27	0.72	0.24
Transitional B cells (CD27- CD38^high^ CD24^high^)	15 [9–23]	13 [8–15]	21 [14–24]	0.06	0.37	0.37	0.07
CD19+ CD21^low^ CD38^low^	6.0 [3.9–7.2]	5.5 [4.0–6.6]	8.0 [4.1–10.5]	0.32	0.97	0.51	0.49
Delay between infection and last vaccine dose							
PI < 90 days	n = 6	n = 3	n = 4				
Leukocytes (CD45+)	14.8 [12.5–24.7]	24.1 [4.3–24.3]	10 [9–12]	0.89	0.89	0.12	0.17
Total B cells (CD19+)	6.26 [4.29–7.50]	11.66 [10.03–14.62]	7.30 [6.28–7.86]	0.07	0.07	0.59	**0.03**
CD27- (CD19+)	66.2 [54.9–76.1]	69.7 [48.2–80.8]	66.4 [58–71]	1.00	1.00	1.00	0.98
CD27+ memory B cells	33.9 [23.7–45.5]	30.3 [18.7–50.8]	33.3 [27–41]	1.00	1.00	1.00	0.98
Switched (CD27+ IgD-IgM-)	37.8 [27.2–53.4]	43.2 [34.4–43.6]	45.1 [39–51]	0.89	0.89	0.89	0.73
Unswitched (CD27+ IgD- IgM-)	15.4 [11.5–19.3]	18.2 [10.2–28.5]	51.2 [44–58]	0.69	0.69	0.69	0.51
DN (CD27- IgD- IgM-)	2.99 [1.80–7.09]	2.69 [1.80–4.47]	3.42 [2.91–5.36]	0.69	0.69	0.69	0.56
Naïve B cells (CD27- IgD+)	60.8 [48.6–76.9]	61.0 [49.6–70.8]	6.2 [58–66]	0.89	0.89	0.89	0.79
Plasmablasts (Switched B cells CD27^high^ CD38^high^)	4.13 [0.00–12.50]	11.95 [6.24–15.88]	3.8 [2.9–6.3]	0.53	0.53	0.74	0.42
Transitional B cells (CD27- CD38^high^ CD24^high^)	14.2 [9.0–18.3]	13.9 [7.1–27.2]	29.4 [23–37]	0.32	1.00	0.12	0.09
CD19+ CD21^low^ CD38^low^	4.95 [3.66–7.17]	5.37 [3.34–8.16]	10.1 [6.1–14.7]	0.56	0.89	0.56	0.50
PI > 90 days	n = 3	n = 9	n = 10				
Leukocytes (CD45+)	21.0 [18.0–21.5]	21.6 [3.3–41.4]	21.6 [10.0–27.3]	1.00	1.00	1.00	0.93
Total B cells (CD19+)	2.96 [2.71–4.49]	10.1 [4.85–13.36]	10.6 [6.48–11.31]	0.76	0.06	**0.05**	**0.04**
CD27- (CD19+)	42.2 [41.8–63.0]	64.3 [59.0–78.2]	59.9 [54.4–76.0]	0.28	0.28	0.28	0.20
CD27+ memory B cells	57.1 [36.5–57.8]	35.6 [21.5–40.5]	39.2 [20.6–43.7]	0.30	0.22	0.22	0.15
Switched (CD27+ IgD- IgM-)	43.0 [39.0–48.0]	59.1 [46.7–63.5]	43.2 [37.4–46.2]	**0.02**	0.20	0.93	**0.02**
Unswitched (CD27+ IgD+ IgM+)	24.4 [18.5–31.8]	13.2 [7.2–14.7]	14.7 [11.3–22.9]	0.20	0.20	1.00	**0.16**
DN (CD27- IgD- IgM-)	4.30 [2.07–7.86]	5.30 [2.38–10.02]	3.86 [2.24–5.32]	0.86	0.86	0.93	0.58
Naïve B cells (CD27- IgD+)	39.7 [33.6–60.6]	54.2 [49.0–76.5]	57.7 [49.6–76.6]	0.96	0.22	0.22	0.24
Plasmablasts (Switched B cells CD27^high^ CD38^high^)	8.0 [6.04–14.40]	11.5 [6.67–23.84]	6.7 [4.92–12.66]	0.71	0.71	0.71	0.49
Transitional B cells (CD27- CD38^high^ CD24^high^)	25.6 [6.5–36.6]	12.8 [7.6–15.8]	19.0 [13.5–22.3]	0.86	0.86	0.93	0.58
CD19+ CD21^low^ CD38^low^	6.31 [4.43–7.94]	5.60 [3.97–6.75]	7.87 [3.59–9.37]	0.68	0.68	0.80	0.58

^1^ Continuous variables are expressed as the median of relative frequency % [IQR]. ^2^ Kruskal–Wallis rank sum test. ^3^ False discovery rate correction for multiple testing.

**Table 4 vaccines-12-00230-t004:** T cells maturation frequencies stratified for SARS-CoV-2 VOCs.

		Variant of Concern VOCs			Multiple Comparisons’ *p* Value ^3^				
% T Cells Subset	Population of Reference	VOC_P_ N = 9 ^1^	VOC_D_, N = 12 ^1^	VOC_O_, N = 14 ^1^	VOC_D_ vs. VOC_O_	VOC_D_ vs. VOC_P_	VOC_O_ vs. VOC_P_	*p* Value ^2^	Adj. *p* Value ^3^
CD3+	CD45+	8 [7–15]	25 [14–32]	21 [13–28]	0.537	**0.043**	0.095	**0.039**	0.126
CD4+	CD3+	36 [27–48]	52 [45–59]	54 [42–58]	0.979	0.055	0.045	**0.043**	0.126
CD8+	CD3+	50 [42–60]	35 [31–46]	35 [29–41]	0.487	**0.038**	**0.038**	**0.023**	0.084
TCM-CD8+	CD8+	7 [5–8]	12 [10–16]	11 [7–14]	0.554	0.074	0.109	0.059	0.151
TEMRA-CD8+	CD8+	33 [17–38]	25 [15–33]	14 [7–26]	0.175	0.374	**0.054**	**0.044**	0.126
TEM-CD8+	CD8+	54 [46–63]	43 [40–53]	51 [38–60]	0.625	0.329	0.625	0.365	0.486
TN-CD8+	CD8+	7 [2–15]	16 [11–22]	18 [11–32]	0.487	0.105	0.105	0.071	0.151
CD8+ (CD57-/PD1-)	CD8+	27 [25–36]	44 [31–53]	46 [39–70]	0.52	0.165	0.164	0.760	0.829
CD8+ (CD57-/PD1+)	CD8+	13 [10–17]	22 [17–34]	16 [9–19]	0.127	0.076	0.825	0.456	0.543
CD8+ (CD57+/PD1-)	CD8+	39 [19–49]	19 [11–26]	29 [10–35]	0.52	0.165	0.165	0.966	0.966
CD8+ (CD57+/PD1+)	CD8+	14 [11–19]	9 [7–13]	7 [3–8]	0.142	0.189	**0.039**	**0.633**	0.717
TEM-CD8+ (CD57-/PD1-)	EM-CD8+	31 [24–48]	35 [27–43]	53 [41–67]	0.071	0.804	0.071	0.044	0.126
TEM-CD8+ (CD57-/PD1+)	EM-CD8+	17 [14–31]	30 [18–39]	24 [13–33]	0.589	0.380	0.592	0.377	0.492
TEM-CD8+ (CD57+/PD1-)	EM-CD8+	23 [16–35]	14 [8–29]	14 [5–18]	0.425	0.361	0.218	0.164	0.259
TEM-CD8+ (CD57+/PD1+)	EM-CD8+	21 [17–24]	12 [5–16]	4 [2–5]	**0.006**	0.110	**0.001**	**<0.001**	**0.004**
TEMRA-CD8+ (CD57-/PD1-)	TEMRA-CD8+	13 [10–19]	21 [18–24]	35 [23–49]	0.072	0.110	**0.054**	**0.017**	0.077
TEMRA-CD8+ (CD57-/PD1+)	TEMRA-CD8+	76 [63–84]	46 [40–66]	46 [28–70]	0.857	0.055	0.071	0.054	0.147
TEMRA-CD8+ (CD57+/PD1-)	TEMRA-CD8+	4 [2–5]	7 [5–17]	4 [1–15]	0.259	0.076	0.777	0.103	0.199
TEMRA-CD8+ (CD57+/PD1+)	TEMRA-CD8+	5 [4–11]	14 [10–25]	3 [2–6]	**0.024**	0.233	0.176	**0.020**	0.082
TCM-CD4+	CD4+	55 [46–58]	58 [47–67]	46 [37–49]	**0.033**	0.241	0.061	**0.011**	0.053
TEMRA-CD4+	CD4+	0.64 [0.30–0.89]	0.43 [0.13–0.71]	1.14 [0.32–2.39]	0.546	0.546	0.546	0.442	0.553
TEM-CD4+	CD4+	40 [31–45]	17 [15–30]	32 [23–41]	0.329	0.128	0.329	0.119	0.217
TN-CD4+	CD4+	10 [4–14]	17 [14–21]	21 [17–32]	0.129	**0.031**	**0.001**	**0.001**	**0.008**
CD4+ (CD57-/PD1-)	CD4+	67 [63–69]	59 [50–66]	88 [78–94]	**0.003**	0.082	**0.045**	**0.001**	**0.013**
CD4+ (CD57-/PD1+)	CD4+	25 [21–28]	33 [27–36]	7 [3–10]	**<0.001**	**0.014**	**0.014**	**<0.001**	**0.004**
CD4+ (CD57+/PD1-)	CD4+	4.5 [2.7–5.1]	1.1 [0.8–2.4]	1.2 [0.4–2.0]	0.537	0.074	0.109	0.067	0.151
CD4+ (CD57+/PD1+)	CD4+	6.4 [2.6–6.7]	4.0 [2.3–8.0]	2.1 [1.2–3.1]	0.125	0.804	0.125	0.103	0.199
TCM-CD4+ (CD57-/PD1-)	CM-CD4+	81 [78–83]	68 [66–74]	80 [64–83]	0.158	**0.002**	0.637	**0.012**	0.069
TCM-CD4+ (CD57-/PD1+)	CM-CD4+	1.33 [0.95–1.45]	0.54 [0.41–0.59]	0.60 [0.40–1.24]	0.456	0.023	0.061	**0.020**	0.082
TCM-CD4+ (CD57+/PD1-)	CM-CD4+	17 [13–20]	31 [25–32]	19 [16–31]	0.102	**0.001**	0.329	**0.004**	**0.03**
TCM-CD4+ (CD57+/PD1+)	CM-CD4+	1.20 [1.01–1.87]	0.90 [0.62–1.08]	0.64 [0.53–1.26]	0.607	0.425	0.406	0.407	0.508
TEM-CD4+ (CD57-/PD1-)	EM-CD4+	58 [45–63]	36 [19–40]	66 [55–81]	**0.002**	**0.021**	0.156	**0.001**	**0.012**
TEM-CD4+ (CD57-/PD1+)	EM-CD4+	22 [5–34]	52 [43–58]	8 [1–14]	**<0.001**	**0.004**	0.156	**<0.001**	**0.004**
TEM-CD4+ (CD57+/PD1-)	EM-CD4+	14 [7–21]	1 [1–6]	13 [9–25]	**0.005**	**0.054**	0.682	**0.005**	**0.035**
TEM-CD4+ (CD57+/PD1+)	EM-CD4+	5 [4–10]	9 [3–22]	3 [2–5]	0.143	0.512	0.263	0.101	0.199

^1^ Continuous variables are expressed as the median of relative frequency % [IQR]. ^2^ Kruskal–Wallis rank sum test. ^3^ False discovery rate correction for multiple testing.

## Data Availability

The de-identified dataset containing demographic information and experimental data is available upon request in the Zenodo repository at https://zenodo.org/records/10204325.

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
