# Peer review of "Changes in the Adaptive Cellular Repertoire after Infection with Different SARS-CoV-2 VOCs in a Cohort of Vaccinated Healthcare Workers"

_vaccines, 2024, doi:10.3390/vaccines12030230_

Round 1
Reviewer 1 Report
Comments and Suggestions for Authors
The manuscript compiles information on adaptive antibody and B- and T-cell immunity to different variants of Covid-19 in health care workers who were vaccinated with 2 and 3 doses of vaccine. The data obtained are unique but functional cell stimulation assays are lacking. In general, clinical data on the symptoms produced by the variants were known. Some points need to be corrected or discussed prior to publication of the manuscript.
- In my opinion, I think the IgM titres were low because they were measured when the IgM had changed to the IgG isotype.
- Fig1c. VOCD and VOCO showed an increase in IgG-RBD-S levels between PV and PI, while VOCP did not change. Vaccine protein S should be more similar to that of VOCP than to that of VOCD and VOCO. How do the authors explain these data?
- Line 84, please define HCW
- Line 144, please remove asterisk.
- Figure S1a. please correct the VOC names in the timeline.
Author Response
Dear Reviewer,
We have carefully read all the comments received and would like to thank you for all your suggestions and constructive comments, which allow us to improve our manuscript.
The manuscript compiles information on adaptive antibody and B- and T-cell immunity to different variants of Covid-19 in health care workers who were vaccinated with 2 and 3 doses of vaccine. The data obtained are unique but functional cell stimulation assays are lacking. In general, clinical data on the symptoms produced by the variants were known. Some points need to be corrected or discussed prior to publication of the manuscript.
- In my opinion, I think the IgM titres were low because they were measured when the IgM had changed to the IgG isotype.
This is an interesting point. The dynamics of IgM elicited by SARS-CoV-2 infection or vaccination is still a controversial question. It is generally accepted that IgM antibodies provide an early-stage response during viral infections prior to the maturation of the class-switched, high affinity IgG response for long-term immunity and immunological memory. During SARS-CoV-2 infection, antigen (Ag)-specific IgM antibodies can be detected as soon as four days after infection with a peak at around 20 days, while Ag-specific IgG increase around 7 days after infection with a peak at approximately 25 days (Jin Y et al. doi: 10.1016/j.ijid.2020.03.065; Long QX et al. doi: 10.1038/s41591-020-0897-1). Our study time point (about two weeks after infection) seems thus to be near to a possible peak of IgM. But several studies reported that a proportion of patients never develop IgM (Li K et al. doi: 10.1038/s41467-020-19943-y; J Long QX et al. doi: 10.1038/s41591-020-0897-1; Xu X et al. doi: 10.1038/s41591-020-0949-6; Chaudhury S. et al. 10.1371/journal.pone.0252628). In our previous work on 1900 vaccinated HCW we explored the role of IgM and we found that after vaccination about 40% of subjects did not develop IgM (Ruggiero et al. https://doi.org/10.1016/j.ebiom.2022.103888 and Piubelli.et al. (doi.org/10.1016/j.ebiom.2023.104471). We added a sentence in the results section pointing out the presence of low levels of IgM in our patients and that this was observed also in previous literature data (lines 204-206)
- Fig1c. VOCD and VOCO showed an increase in IgG-RBD-S levels between PV and PI, while VOCP did not change. Vaccine protein S should be more similar to that of VOCP than to that of VOCD and VOCO. How do the authors explain these data?
We thank the reviewer for the interesting comment. We agree with the observation that Vaccine S protein is more similar to the VOCP and that this fact could influence IgG-RBD-S production. Therefore the higher similarity of the S protein could result in the production of higher affinity antibodies for the pre-delta variant, which could achieve a more efficient response, even with lower levels. We added a comment in the discussion highlighting this concept. Lines 413-417
- Line 84, please define HCW.
The definition of HCWs Healthcare Workers was already expressed in line 49
- Line 144, please remove asterisk. Done, substituted with the multiplication sign
- Figure S1a. please correct the VOC names in the timeline. Done
Reviewer 2 Report
Comments and Suggestions for Authors
The article entitled Changing in Adaptive Cellular Repertoire after Different 2
SARS-CoV-2 VOCs Infection in a Cohort of Vaccinated Healthcare Workers, is interesting and can be accepted.
However, I have few questions:
1) The study primarily focuses on the functions of T cells and B cells following vaccination. To thoroughly illustrate immune cell functions, it is crucial to measure both pro-inflammatory and anti-inflammatory cytokine levels before and after vaccination. This approach will provide a comprehensive view of immune response enhancement.
2) The decision to express antibody values using an index system rather than providing direct serum levels of antibodies raises a question. While the current method is valid, it would be more insightful to measure serum immunoglobulin titers both before and after vaccination. This would offer a deeper understanding of the efficacy of the vaccination program.
3) It's important to question whether there is any seroconversion occurring as a result of the vaccination.
4) In the context of SARS-CoV-2 infection, the expression and function of follicular T cells are particularly critical due to their role in memory cell development. Therefore, it is essential to analyze the cytokine network and to measure serum antibody levels both before and after vaccination to evaluate the vaccine's impact on these immune components.
5) Introduction should be more specific for better understanding stating the reasons why these studies are important to validate vaccination.
Author Response
Dear Reviewer,
We have carefully read all the comments received and would like to thank you for all your suggestions and constructive comments, which allow us to improve our manuscript.
The article entitled Changing in Adaptive Cellular Repertoire after Different SARS-CoV-2 VOCs Infection in a Cohort of Vaccinated Healthcare Workers, is interesting and can be accepted.
However, I have few questions:
1) The study primarily focuses on the functions of T cells and B cells following vaccination. To thoroughly illustrate immune cell functions, it is crucial to measure both pro-inflammatory and anti-inflammatory cytokine levels before and after vaccination. This approach will provide a comprehensive view of immune response enhancement.
We thank the reviewer for the comment. The main objective of our study was the evaluation of the involvement of the different cellular compartments after SARS-CoV-2 VOCs Infection in vaccinated subjects presenting mild or no symptoms. In fact, we analysed T and B cells subpopulations only after the infection. We think that the timepoint of our experimental design (about two weeks after the infection) should not be suitable and informative for pro-inflammatory and anti-inflammatory cytokine measurements, considering also that all the patients presented very mild or no symptoms, as outlined by Soares-Schanoski at al. (Front Immunol. 2022 Apr 5;13:821730. doi: 10.3389/fimmu.2022.821730). For each subject we also considered the humoral response after the last vaccine dose, only to evaluate the anti-SARS-CoV-2 antibodies levels produced by infection in comparison with those produced after vaccination. We adapted the abstract and the introduction paragraphs to better explain this point and the goal of our work. Lines 16-17; 23-24; 77-84
2) The decision to express antibody values using an index system rather than providing direct serum levels of antibodies raises a question. While the current method is valid, it would be more insightful to measure serum immunoglobulin titers both before and after vaccination. This would offer a deeper understanding of the efficacy of the vaccination program.
We thank the reviewer for the observation. We agree with the reviewer that reporting absolute quantification levels for all the antibody types could be beneficial, but for IgG-N and IgM the used Abbott method provides only a relative quantification using a specific calibrator. However we always reported and discussed data comparing groups, considering qualitative data, as already published by many authors (Gabriel N. Maine, at al. https://doi.org/10.1016/j.jcv.2020.104663; Andrew Bryan et al. https://doi.org/10.1128/jcm.00941-20 and Marina Pollán et. al. https://doi.org/10.1016/S0140-6736(20)31483-5). The serum antibodies levels before and after vaccination for our cohort were extensively described in our previous manuscripts (Buonfrate D. et. al doi: 10.1016/j.cmi.2021.07.024. Epub 2021 Jul 28. , Delle Carbonare doi: 10.1038/s43856-021-00039-7, Ruggiero A. et. al. , doi: 10.1016/j.ebiom.2022.103888., Piubelli, C et.al. doi: 10.1016/j.ebiom.2023.104471), but the objective of the present paper was focused on the changing in the immune response after SARs-CoV-2 challenge. However, we appreciate the reviewer's feedback. As suggested, we have included a Supplementary Figure (S2) in the manuscript to show the antibodies levels both before and after vaccination in our cohort. The figure is cited in results paragraph, Line 201-202
3) It's important to question whether there is any seroconversion occurring as a result of the vaccination.
As reported in the previous point, the seroconversion after vaccination has been extensively described in our previous published manuscripts. Anyway, as reported in table 2, all the subjects presented high levels of IgG-RBD-S with a mean of 2336 (BAU/ml) [IQR; 1198-4495], confirming an efficient seroconversion. This is now more visible with the addition of Supplementary Figure (S2).
4) In the context of SARS-CoV-2 infection, the expression and function of follicular T cells are particularly critical due to their role in memory cell development. Therefore, it is essential to analyze the cytokine network and to measure serum antibody levels both before and after vaccination to evaluate the vaccine's impact on these immune components.
We thank the reviewer for the observation. As highlighted by the reviewer, the vaccine-specific T follicular helper (Tfh) cells are a key component to an effective immune response, but they are prevalently localized in the draining lymph node (LN) and assist in the selection of highly specific B-cell clones for the production of neutralizing antibodies. While responses to vaccination occur in the LN, access to this environment for analysis of the human immune response is difficult and can be invasive. Circulating Tfh cells (cTfh) have been used as a surrogate for LN Tfh cells and are easily measured in the peripheral blood. This subset has been shown to correlate with neutralizing antibody (NAb) levels following various vaccinations, which are key for the prevention of infection and clearance of acute viral infections. As described by Boyd MAA, et.al. doi: 10.1111/imcb.12635. Also Dawei Cui et. al. described T follicular cells following Covid infection and described the functional features and roles of virus-specific Tfh cells in the immunopathogenesis of SARS-CoV-2 infection and in COVID-19 vaccines, and highlight the potential of targeting Tfh cells as therapeutic strategy against SARS-CoV-2 infection. Emerging evidence indicates that functional characterization of Tfh cells and their subsets will provide novel insights into improved vaccine design and therapeutic strategies to prevent and control various viral infections including SARS-CoV-2 infection. https://doi.org/10.3389/fimmu.2021.731100. Despite this, neither of these two studies considered the cytokine profile to better described the Tfh profile.
All subjects involved in our study were healthy, immunocompetent and demonstrated a high antibodies production (Figure 1, Supplementary Figure S2), indicating an efficient humoral component response. In our study we analyze different circulating B supopulations of usnswitched –switched and plasmablasts for understand the impact of different VOC infection into circulating B trained cell compartment, since our goal was to evaluate the state of maturation, activation, and potential senescence or exhaustion of the immune system following exposure to different SARS-CoV-2 VOCs.
To accurately answer at reviewer's question, it may be necessary to conduct in-vitro studies using peptides derived from different VOCs of Sars-Cov-2 and assess the presence of neutralizing antibodies and different T and B memory cells to these epitopes in the serum of vaccinated healthy subjects. However, this study is not feasible for us, since we did not stored PBMCs. We highlighted this point as a limitation of the study in the discussion paragraph (Lines 487-490).
5) Introduction should be more specific for better understanding stating the reasons why these studies are important to validate vaccination.
The objective of the study was to monitor the dynamics and persistence of the immune response versus different SARS-CoV-2 VOCs emerged during the pandemic period, referring to the immunological state after the last vaccine dose before viral infection. We evaluated both the humoral and cell-mediated response following exposure to different SARS-CoV-2 VOCs, by assessing anti-Spike and anti-Nucleocapsid antibodies, as well as characterizing the B and T cells repertoire, including maturation, activation, and potential senescence or exhaustion markers, in a cohort of vaccinated HCWs who experienced breakthrough infection. For each subject we also considered the humoral response after the last vaccine dose, to evaluate the anti-SARS-CoV-2 antibodies levels produced by infection in comparison with those produced after vaccination. As suggested by the reviewer, we adapted the abstract and the introduction paragraphs to better explain the goal of our work. Lines 16-17; 23-24; 77-84
Reviewer 3 Report
Comments and Suggestions for Authors
The study of Calder et al., entitled: “Changing in Adaptive Cellular Repertoire after Different SARS-CoV-2 VOCs Infection in a Cohort of Vaccinated Healthcare Workers” monitored the dynamics and persistence of the immune response versus different SARS-CoV-2 variants emerged during the pandemic period (2021-2022), in a cohort of vaccinated health care workers, who experienced breakthrough infection in the Pre-Delta, Delta, and Omicron waves. The authors evaluated both the humoral and cellular immune response, referring them to the immunological state after the last vaccine dose before infection.
The presented results highlight that the immune response against the Delta variant of concern (VOC) involved mainly an adaptive humoral and switched memory B cells component, more than 3 months after the vaccine dose, showing a high percentage of depleted adaptive T cells. Omicron infections triggered a consistent production of non-vaccine-associated anti-N antibodies, for balancing the spike epitope immune escape mechanisms.
The experiments seem to be well-performed.
The manuscript should be accepted for publication after minor corrections:
On page 2, line 91 authors mentioned “From March 2021 to June 2022, 36 HCWs were followed up”, while in Table 1 it is quoted All n=35.
Please harmonize the number of persons involved in the study.
Also in Table 1: Subjects infected after 90 days from last vaccine doses; n (%)
VOCP 3 VOCD 9 VOCO 10 All 23
Please check if was it 22 or 23.
Page 4, line 162, could you please elaborate on setting 10.000 events in the CD3+ gate for the T cells panel and 1.000 events in the CD19+ gate for the B cells panel?
Page 5, line 190, 192: it would be more convenient to use either more or within 90 days, or more or within three months.
Page 12, line 324, please correct the subtitle.
